# C1q/TNF-Related Proteins 1, 6 and 8 Are Involved in Corneal Epithelial Wound Closure by Targeting Relaxin Receptor RXFP1 In Vitro

**DOI:** 10.3390/ijms24076839

**Published:** 2023-04-06

**Authors:** Hagen Fabian Nicolaus, Thomas Klonisch, Friedrich Paulsen, Fabian Garreis

**Affiliations:** 1Institute of Functional and Clinical Anatomy, Friedrich-Alexander-Universität Erlangen-Nürnberg, 91054 Erlangen, Germany; 2Institute of Experimental and Clinical Pharmacology and Toxicology, Friedrich-Alexander-Universität Erlangen-Nürnberg, 91054 Erlangen, Germany; 3Universitätsklinikum Erlangen, Friedrich-Alexander-Universität Erlangen-Nürnberg, 91054 Erlangen, Germany; 4Department of Human Anatomy and Cell Science, Rady Faculty of Health Sciences, College of Medicine, Winnipeg, MB R3E 0J9, Canada; 5Department of Pathology, Rady Faculty of Health Sciences, College of Medicine, University of Manitoba, Winnipeg, MB R3E 3P5, Canada; 6Department of Medical Microbiology & Infectious Diseases, Rady Faculty of Health Sciences, College of Medicine, University of Manitoba, Winnipeg, MB R3E 0J9, Canada; 7Research Institute in Oncology and Hematology (RIOH), Cancer Care Manitoba, Winnipeg, MB R3E 0J9, Canada

**Keywords:** CTRP1, CTRP6, CTRP8, C1q/TNF-related proteins, C1QTNF, RXFP1, relaxin receptor, corneal wound closure, ocular surface, lacrimal apparatus

## Abstract

Inadequate wound healing of ocular surface injuries can lead to permanent visual impairment. The relaxin ligand-receptor system has been demonstrated to promote corneal wound healing through increased cell migration and modulation of extracellular matrix formation. Recently, C1q/tumor necrosis factor-related protein (CTRP) 8 was identified as a novel interaction partner of relaxin receptor RXFP1. Additional data also suggest a role for CTRP1 and CTRP6 in RXFP1-mediated cAMP signaling. However, the role of CTRP1, CTRP6 and CTRP8 at the ocular surface remains unclear. In this study, we investigated the effects of CTRP1, CTRP6, and CTRP8 on epithelial ocular surface wound closure and their dependence on the RXFP1 receptor pathway. CTRP1, CTRP6, and CTRP8 expression was analyzed by RT-PCR and immunohistochemistry in human tissues and cell lines derived from the ocular surface and lacrimal apparatus. In vitro ocular surface wound modeling was performed using scratch assays. We analyzed the effects of recombinant CTRP1, CTRP6, and CTRP8 on cell proliferation and migration in human corneal and conjunctival epithelial cell lines. Dependence on RXFP1 signaling was established by inhibiting ligand binding to RXFP1 using a specific anti-RXFP1 antibody. We detected the expression of CTRP1, CTRP6, and CTRP8 in human tissue samples of the cornea, conjunctiva, meibomian gland, efferent tear ducts, and lacrimal gland, as well as in human corneal, conjunctival, and meibomian gland epithelial cell lines. Scratch assays revealed a dose-dependent increase in the closure rate of surface defects in human corneal epithelial cells after treatment with CTRP1, CTRP6, and CTRP8, but not in conjunctival epithelial cells. Inhibition of RXFP1 fully attenuated the effect of CTRP8 on the closure rate of surface defects in human corneal epithelial cells, whereas the CTRP1 and CTRP6 effects were not completely suppressed. Conclusions: Our findings demonstrate a novel role for CTRP1, CTRP6, and CTRP8 in corneal epithelial wound closure and suggest an involvement of the relaxin receptor RXFP1 signaling pathway. This could be a first step toward new approaches for pharmacological and therapeutic intervention.

## 1. Introduction

Damage of the ocular surface represents a common and increasingly important problem in the clinical routine of ophthalmologists [1,2]. Causes of ocular surface damage can be versatile and include trauma, infection, refractory surgery but also chronic surface damage due to inflammation and dry eye disease. To preserve the physiological barrier function as well transparency for refractive functions of the cornea, effective wound healing is key. At the ocular surface, this highly complex process is mediated by a plethora of immunological factors, growth factors, and other hormones [1,2]. This includes relaxin 2 (RLN2), insulin-like factor 3 (INSL3), and their cognate receptors relaxin/insulin-like family peptide receptor 1 and 2 (RXFP1/2), which are able to enhance corneal wound repair at the ocular surface and protect against permanent vision impairment through chronic ulcera and corneal fibrosis [3,4]. Recently, the C1q/tumor necrosis factor-related protein (CTRP) 8 was identified as a novel ligand of RXFP1 that is capable of activating several RXFP1-mediated signaling pathways supporting enhanced cell motility and survival [5].

C1q/tumor necrosis factor-related proteins (CTRPs) are a group of secretory peptides that share a common structure composed of four domains: (1) a signaling peptide, (2) a short variable region, (3) a collagenous domain, and (4) a globular C1q domain. Unlike adiponectin, the most studied member of the CTRP family, which is almost exclusively expressed in adipocytes, the remaining 16 CTRPs (CTRP1-CTRP9, 9B, 10–15) are widely expressed in various tissues and cell types [6,7,8,9]. CTRP1/6/8 share the same evolutionary clade, a high sequence homology in the putative RXFP1 binding peptide site and can elicit cAMP signaling in RXFP1 expressing cells [5,10].

CTRP1 is expressed in the heart, kidney, liver, and predominantly in the adipose tissue and was shown to modulate metabolic and cardiovascular function [6,8,11,12,13,14]. Expression of CTRP1 can be induced by interleukin (IL) 1β and tumor necrosis factor α (TNF-α) in adipocytes and is closely linked to inflammatory processes and elevated serum levels of pro-inflammatory cytokines, such as IL-6 [14,15,16,17,18]. Furthermore, CTRP1 was recently found to promote cell proliferation and migration as well as tumor invasion [19,20,21].

CTRP6 expression has been described among others in the adipose tissue, placenta, lung, and heart [6,8]. Besides its involvement in cardiovascular diseases and fat metabolism [6,22,23], CTRP6 plays an important role in inflammation as it acts as an inhibitor of the alternative pathway of the complement system and increases IL-10 expression in macrophages [24,25]. In addition, CTRP6 inhibits transforming growth factor β1 (TGF-β1)-induced fibrosis in dermal, renal, and cardiac fibroblasts [26,27,28,29]. Moreover, CTRP6 is involved in cell proliferation and tumor genesis. However, the effects of CTRP6 appear to differ widely depending on tissue and cell type [30,31,32].

CTRP8 is the least understood member of the CTRP family, mainly because there is no orthologue of the CTRP8 gene in the mouse genome [9]. Recently, RXFP1 was identified as a receptor for CTRP8, where the N-terminal “YAAFSVG” peptide motif in the globular C1q domain of CTRP8 can facilitate RXFP1-mediated cAMP signaling [5]. The CTRP8-RXFP1 ligand-receptor system has been shown to activate different signaling pathways to promote glioblastoma cell migration by modulating the actin cytoskeleton [33].

Our group previously demonstrated the expression of RXFP1 at the ocular surface and the significance of RLN2-RXFP1 interaction for ocular surface wound healing [3]. Activation of RXFP1 by RLN2 promotes corneal re-epithelialization mainly through increased cell migration. In addition, extracellular matrix (ECM) formation is modulated by RLN2-RXFP1 signaling through altered collagen synthesis and enhanced matrix metalloproteinases (MMPs) and tissue inhibitors of matrix metalloproteinases (TIMPs) expression [3].

Currently, the expression of CTRP1, CTRP6, and CTRP8 at the ocular surface and their role in ocular surface wound healing have not been investigated. In the present study, we characterized the expression of CTRP1, CTRP6, and CTRP8 in tissues and cell lines of the ocular surface and lacrimal apparatus. In addition, in vitro wound closure assays provided first information on the role of CTRP1, CTRP6, and CTRP8 in the recovery of ocular surface epithelial wounds and gave insight into the involvement of the RXFP1 signaling pathway in this clinically important process. Our study adds further relevance to CTRP1/6/8-RXFP1 biology and is a first step toward possible new pharmaceutical and therapeutic approaches for the treatment of ocular surface wounds.

## 2. Results

### 2.1. Gene Expression of CTRP1, CTRP6, and CTRP8 in Tissues of the Lacrimal Apparatus and Ocular Surface

CTRP1, CTRP6, and CTRP8 gene expression in human tissue and cell line of the ocular surface and lacrimal apparatus was investigated by RT-PCR. CTRP1 (399 bp)- and CTRP6 (400 bp)-specific cDNA amplification products were detected in human cornea, conjunctiva, lacrimal gland, and nasolacrimal ducts as well as cell lines of human corneal epithelium (HCE and hTCEpi), conjunctival epithelium (HCjE and IOBA-NHC), and meibomian gland epithelial cells (HMGEC) (Figure 1). Specific CTRP8 amplicons (350 bp) were only detectable in human corneal epithelial cell line (HCE) and IOBA-normal human conjunctiva cell line (IOBA-NHC), with variable expression levels throughout the investigated samples. There was no expression of CTRP8 transcripts in hTCEpi, HCjE, HMGEC, and human cornea (Appendix A). Negative signals in the control reactions resulting from omission of template cDNA confirmed the absence of genomic DNA. cDNAs from liver (CTRP1), placenta (CTRP6), or testis (CTRP8) served as the positive control (PC) and were used to validate the identity of anticipated CTRP1/6/8 PCR products.

### 2.2. CTRP1, CTRP6, and CTRP8 Protein Expression in Tissues of the Lacrimal Apparatus and Ocular Surface

To verify gene expression of CTRP1, CTRP6, and CTRP8, immunohistochemical analysis of formalin-fixed paraffin-embedded tissue sections from human cadavers as well as formalin-fixed cell lines was performed. Figure 2 shows CTRP protein expression in tissues of the ocular surface and lacrimal apparatus. Immunoreactive CTRP1, CTRP6, and CTRP8 were present (peri-)nuclear and in the cytoplasm of all layers of the corneal epithelium. Moreover, CTRP1 and CTRP6 revealed some cytoplasmatic and perinuclear vesicular formation. To eliminate staining artifacts and to verify CTRP1 distribution, antibody specificity was confirmed by negative staining after elimination of antibody reactivity by antigen pre-adsorption (Appendix A). Fibroblasts of the corneal stroma and endothelial cells also revealed reactivity to the three investigated CTRPs. Conjunctival epithelium showed cytoplasmatic immunoreactivity to CTRP1, CTRP6, and CTRP8. However, the stored secretion products of intraepithelial goblet cells did not react (CTRP1 and CTRP6) or only showed weak immunoreaction (CTRP8). In the meibomian glands, CTRP1, CTRP6, and CTRP8 were positive in meibocytes, especially in basal and mature meibocytes, whereas hypermature and apoptotic meibocytes showed weaker reactivity. Strong CTRP8 reactivity in particular could be observed in basal meibocytes. Moreover, lining cells of the excretory duct system showed positive staining for CTRP1, CTRP6, and CTRP8. In the lacrimal gland, acinar cells and cells of the tubular system demonstrated immunoreactivity with all three investigated CTRPs. In the nasolacrimal ducts, epithelial cells of the lacrimal sac and nasolacrimal duct demonstrated strong CTRP1, CTRP6, and CTRP8 antibody immunoreactivity. Moreover, throughout the investigated tissue sections single cells within the loose connective tissue and subepithelial single cells, endothelium of small blood vessels, and epithelium of the epidermis showed positive CTRP1, CTRP6, and CTRP8 antibody immunoreactivity. Immunohistochemical studies of HCE, HCjE, and HMGEC (undifferentiated and differentiated) revealed localization of CTRP1, CTRP6, and CTRP8 in the cytoplasm and (peri-)nuclear (Figure 3). Control sections with non-immune IgG were consistently negative for each of the investigated tissues and cell lines. Tissue sections used as positive controls verified the immunodetection method for each of the CTRP antibodies employed (Appendix A).

### 2.3. Dose-Dependent Impact on Ocular Surface Epithelial Wound Closure

In vitro epithelial wound closure assays revealed dose-dependent effects of CTRP1, CTRP6, and CTRP8. HCE cells stimulated with 100 ng/mL CTRP1 showed a significantly decreased surface defect area compared to the control after 12 h of treatment, remaining significant for the whole observation period (Figure 4). However, 10 ng/mL CTRP1 did not show a significant reduction in the surface defect area compared to the control (Appendix A). The surface defect closure rate increased significantly (1.34 ± 0.14-fold; *p* = 0.02) in the presence of 100 ng/mL CTRP1, while 10 ng/mL CTRP1 did not increase the surface defect closure significantly.

Like CTRP1, the remaining surface defect area of HCE cells stimulated with 100 ng/mL CTRP6 was significantly smaller when compared to the control, starting at 6 h of treatment and the remaining observation period (Figure 5). Moreover, the surface defect closure rate was significantly increased by treatment of cells with 100 ng/mL CTRP6 (1.31 ± 0.10-fold; *p* = 0.03). Stimulation with 10 ng/mL CTRP6 showed no significant change to the surface defect area and the surface defect closure rate compared to the control (Appendix A).

The stimulation of HCE cells with 100 ng/mL CTRP8 also led to a significantly decreased remaining surface defect area at 6, 12, 24, and 36 h of stimulation but lost significance at 48 h (Figure 6). On the other hand, stimulation with 10 ng/mL CTRP8 did not show a significant effect on the surface defect area compared to the control at any time point of observation (Appendix A). Treatment with CTRP8 did not significantly affect the surface defect closure rate in HCE cells. However, we observed a positive trend in the surface defect closure rate in cells stimulated with 100 ng/mL of CTRP8 (1.26 ± 0.17-fold; *p* = 0.23).

HCjE cells showed no significant increase in the surface defect closure rate when stimulated with 10 or 100 ng/mL of CTRP1, CTRP6 or CTRP8 when compared to the control (Figure 7).

Based on these positive results for CTRP1, CTRP6, and CTRP8 on the remaining surface defect area and the positive trend on the surface defect closure rate in HCE cells, we hereinafter moved forward to lager-scaled corneal wound closure assays to detect a possible CTRP-RXFP1 interaction involved in CTRP1/6/8 mechanism of action.

### 2.4. Dependence on RXFP1 Receptor Pathway

We further analyzed the effects of CTRP1, CTRP6, and CTRP8 on corneal epithelial wound closure by determining whether these effects were dependent on functional RXFP1 receptor. HCE were stimulated with 100 ng/mL CTRP1, CTRP6, or CTRP8 in the presence or absence of a specific inhibitory anti-RXFP1 antibody in additional in vitro epithelial wound closure assays. RXFP1 protein expression in HCE cells was confirmed prior to treatment by immunohistochemical staining (Figure 8). The inhibition of RXFP1 by the antibody was verified by stimulation of HCE cells with 100 ng/mL relaxin-2 (RLN2), which served as the positive control with or without the antibody. Stimulation with RLN2 led to a significantly increased surface defect closure rate compared to the control (1.40 ± 0.06-fold; *p* < 0.001). The presence of the anti-RXFP1 antibody fully diminished the effects of RLN2 on the surface defect closure rate to control levels (0.99 ± 0.04-fold). Treatment with the antibody alone, however, did not affect the surface defect closure rate significantly (0.96 ± 0.06-fold; *p* = 0.51).

HCE cells showed a significant increase in the surface defect closure rate when treated with either CTRP1 (1.41 ± 0.08-fold; *p* < 0.001), CTRP6 (1.20 ± 0.09-fold; *p* = 0.01), or CTRP8 (1.24 ± 0.05-fold; *p* = 0.002), respectively, compared to the untreated control (Figure 9). In presence of the anti-RXFP1 antibody, the surface defect closure rate in CTRP1 treated cells was reduced (1.29 ± 0.08-fold; *p* = 0.80) compared to cells treated with CTRP1 alone, but it remained significantly increased compared to the untreated control (*p* < 0.001). In CTRP6-treated cells, the presence of the anti-RXFP1 antibody reduced the surface defect closure rate to control levels (1.02 ± 0.05-fold; *p* > 0.99). However, when compared to cells treated with CTRP6 alone, this reduction in the surface defect closure rate did not reach statistical significance (*p* = 0.18). RXFP1-inhibition in CTRP8-treated cells resulted in a significantly reduced surface defect closure rate (0.99 ± 0.04-fold; *p* = 0.02) compared to cells treated with CTRP8 alone, with no significant difference to control levels (*p* > 0.99).

## 3. Materials and Methods

### 3.1. Subjects and Tissues

Lacrimal glands, upper eyelids, conjunctivas, corneas, and nasolacrimal ducts (consisting of lacrimal sac and nasolacrimal duct) were obtained from human cadavers donated by written testamentary disposition and in accordance with German law to the Institute of Functional and Clinical Anatomy of Friedrich-Alexander-Universität Erlangen-Nürnberg (FAU), Germany. All tissues were dissected from the cadavers within 4 to 12 h of death. Donors were free of recent trauma, eye and nasal infections, and diseases involving or affecting lacrimal apparatus or ocular surface function. After dissection, tissues from the eye of each cadaver were prepared for paraffin-embedding and were fixed in 4% paraformaldehyde. Tissues for molecular biological investigations were immediately frozen at −80 °C.

### 3.2. Cells and Cell Culture

SV40-transformed human corneal epithelial cells (HCE cells; obtained from Kaoru Araki-Sasaki, Tane Memorial Eye Hospital, Osaka, Japan) [34] as well as immortalized conjunctival epithelial cells (HCjE) [35] were cultured as monolayer and used for scratch assays. HCE cells were cultured in Dulbecco modified Eagle medium (DMEM/HAMs F12 1:1; PAA Laboratories GmbH, Pasching, Austria) containing 10% fetal calf serum (FCS; Biochrom AG, Berlin, Germany). HCjE cells were cultured in keratinocyte serum-free medium (KSFM; without CaCl_2_) (Gibco^TM^ ThermoFisher Scientific, Waltham, MA, USA) supplemented with 25 µg/mL Bovine Pituitary Extract (BPE), 5 ng/mL of epidermal growth factor (EGF) (all supplied with the medium), 0.5 mL of 0.3 M CaCl_2_ (Sigma-Aldrich, Darmstadt, Germany), and 1% penicillin/streptomycin (Sigma-Aldrich, Darmstadt, Germany). For scratch assays, HCE and HCjE cells were seeded in six well plates and incubated in a humidified incubator containing 5% CO_2_ at 37  °C. Reaching 80–90% confluence, medium was changed in HCjE cells to serum-containing medium composed of DMEM/HAMs F12 1:1, 10% FCS, 10 ng/mL EGF (Sigma-Aldrich, Darmstadt, Germany), and 1% penicillin/streptomycin to induce differentiation of HCjE cells. At full confluence and before treatment, cells were washed in phosphate-buffered saline (PBS) and incubated in serum-free medium (DMEM/HAMS F-12 1:1) for 3 h. Additionally, human telomerase-immortalized corneal epithelial cells (hTCEpi; obtained from James Jester, University of California, Irvine, CA, USA) [36] and human spontaneously immortalized epithelial cell line from normal human conjunctiva (IOBA-NHC; obtained from Yolanda Diebold, University Institute of Applied Ophthalmobiology (IOBA), University of Valladolid, Valladolid, Spain) [37] were cultured as a monolayer as described before [38,39] and used for further investigations concerning the corneal and conjunctival epithelium, respectively. Human Meibomian gland epithelial cell line (HMGEC; obtained from David Sullivan, Schepens Eye Research Institute, Boston, MA, USA) was cultured with and without 10% fetal calf serum to initiate differentiation as described previously [40].

### 3.3. RNA Preparation and Complementary DNA (cDNA) Synthesis

For conventional reverse transcriptase-polymerase chain reaction (RT-PCR), tissue samples of the lacrimal gland, nasolacrimal ducts, cornea, and conjunctiva (*n* = 3, respectively) were crushed in an agate mortar under liquid nitrogen and homogenized with Speedmill Plus (Analytik Jena AG, Jena, Germany). Insoluble material was removed by centrifugation (12,000× *g*, 5 min, 4 °C). Total RNA was extracted from the samples using the Rneasy^®^ Mini Kit (Qiagen, Hilden, Germany). In addition, total RNA was isolated from cultivated HCE, hTCEpi, HCjE, IOBA-NHC, and HMGEC (differentiated and undifferentiated) cell lines by use of the peqGOLD Trifast^TM^ reagent (PeqLab, Erlangen, Germany) according to the manufacturer’s protocol. Crude RNA was purified with isopropanol and repeated ethanol precipitation, and contaminated DNA was destroyed by digestion with Rnase-free Dnase I for 30 min at 37 °C (ThermoFisher Scientific, Waltham, MA, USA). The Dnase was heat-denatured for 10 min at 65 °C. Reverse transcription of all RNA samples to first strand cDNA was performed by using the RevertAid^TM^ Reverse Transcriptase Kit (ThermoFisher Scientific, Waltham, MA, USA) according to the manufacturer’s protocol. For each reaction, 2 µg of total RNA and 10 pmol oligo (dT)_18_ primer (Fermentas) was used. To assess integrity of the transcribed cDNA, the ubiquitously expressed β-actin served as the internal control.

### 3.4. Reverse Transcriptase-Polymerase Chain Reaction (RT-PCR)

CTRP1 and CTRP6 transcripts were amplified using the ThermoFisher Scientific Kit (Waltham, MA, USA) according to the manufacturer’s protocol. Each reaction was prepared with 2 μL cDNA, 11.8 μL H_2_O, 2 μL 50 mM MgCl_2_, 1 μL dNTP, 2 μL 10 × PCR buffer, 0.2 μL (5 U/µL) Taq DNA polymerase (Invitrogen, Karlsruhe, Germany), and 0.5 μL (100 pmol) of each of the following primers: CTRP1 sense 5′-ACC GCC GTG CCC CAG ATC AAC-3′, antisense 5′-CAC CAC CTC CTC CTC GTT CTT C-3′ and CTRP6 sense 5′-ATG GTG GAG CTC ACC TTT GAC A-3′, antisense 5′-AGC ACC CAT CAA GGT TCA CA-3′. Reaction underwent an initial cycle at 95 °C for 3 min followed by 40 cycles of 95 °C for 60 s, primer specific annealing temperature (CTRP1: 64 °C; CTRP6: 68 °C) for 60 s, 72 °C for 120 s, and a final elongation at 72 °C for 5 min. CTRP8 transcripts were amplified by use of the Q5^®^ High-Fidelity DNA-Polymerase Kit (New England BioLabs, Ipswich, MA, USA) according to the manufacturer’s protocol. Each PCR reaction contained 2 µL cDNA, 9.75 µL H_2_O, 0.5 µL dNTP, 5 µL 5 × Q5^®^ Reaction Buffer, 5 µL 5 × Q5^®^ High GC Enhancer, 0.25 µL (2 U/µL) Q5^®^ High-Fidelity DNA-Polymerase and 1.25 µL (10 pmol) of each of the following primers: CTRP8 sense 5′-ACG GCC CAC TAT AGA CAT CGA A-3′, antisense 5′-TGT AGT TCC AGG TGT GCA CGT T-3′. Reaction underwent an initial cycle at 98 °C for 30 s followed by 40 cycles of 98 °C for 10 s, primer specific annealing temperature (CTRP8: 66 °C) for 30 s, 72 °C for 30 s, and a final elongation at 72 °C for 120 s. All primers were synthesized at Metabion (Planegg/Steinkirchen, Germany). Ten microliters of the PCR were loaded onto a 1.5% agarose gel and the amplified products were visualized via fluorescence after electrophoresis (CTRP1: 399 bp; CTRP6: 400 bp; CTRP8: 350 bp). For each investigated tissue and cell line at least three individual experiments were performed (*n* ≥ 3).

### 3.5. Immunohistochemistry

For analysis by immunohistochemistry, lacrimal glands, nasolacrimal ducts, cornea, and upper eye lids with conjunctiva and meibomian glands from human cadaver were fixed in paraformaldehyde (PFA), embedded in paraffin, sectioned, and dewaxed by xylol to descending alcohol series as described previously [41]. Additionally, HCE, HCjE and HMGEC cells were seeded in slides (Nunc Lab-Tek™ Chamber Slides, ThermoFisher Scientific, Waltham, MA, USA) and fixed in 4% PFA at 50% confluence. Immunohistochemical staining was performed with polyclonal rabbit anti-CTRP1 (1:50–1:100; ab25973, abcam, Cambridge, UK), rabbit anti-CTRP6 (1:20–1:50; AS-54561, AnaSpec, Fremont, CA, USA), rabbit anti-CTRP8 (1:100–200; STJ92519, St. John’s Laboratory, London, UK), and mouse anti-RXFP1 (1:300, H00059350-M01, Abnova, Taipei, Taiwan) antibodies. Sections were treated with 3% hydrogen peroxide for 10 min, followed by boiling in 10 mM citrate buffer (pH 6) for 10 min for antigen retrieval. Non-specific binding was inhibited by incubation with 5% secondary antibody–specific normal serum (Dako, Santa Clara, CA, USA) in Tris-buffered saline with Tween 20 (TBST). The sections were incubated overnight with the primary antibody at 4 °C and with the secondary antibodies (1:200) at room temperature for 2 h. Visualization was achieved with horseradish peroxidase-labeled streptavidin-biotin complex (StreptABComplex/HRP; Dako) and 3-amino-9-ethylcarbazole (AEC; Dako). Sections were counterstained with hemalum and mounted in Entellan (Dako). Sections of the liver and kidney (CTRP1), placenta (CTRP6 and RXFP1), and testis (CTRP8) served as positive controls. Control sections were incubated with non-immune IgG instead of primary antibody to determine possible non-specific binding of the IgG. Furthermore, CTRP1 antibody specificity was confirmed by elimination of antibody reactivity with antigen pre-adsorption with 3 μg peptide/μg antibody (Human CTRP1 Peptide, ab39751, abcam, Cambridge, UK). All slides were examined with a Keyence BZ 9000 microscope.

### 3.6. Wound Closure Assay (Scratch Assay)

In vitro ocular surface epithelial wound closure modeling was performed using a scratch assay. HCE and HCjE cells were grown in six well plates as mentioned above until full confluence was reached. The cell layer in each well was scratched three times with a 200 µL plastic pipette tip creating a cell-free area (i.e., a ‘wound’) of similar width. Cells were washed twice with PBS to remove debris, and fresh serum-free medium was added. Images of a representative wound area of each scratch were taken (Zeiss Axiovert 40 CFL; Leica MC170 HD microscope camera) and the area was marked for follow-up observation. Cells were then stimulated with different concentrations (10 ng/mL or 100 ng/mL) of recombinant human CTRP1 (#00083-01-10, Aviscera Bioscience, Santa Clara, CA, USA), CTRP6 (#00089-02-10, Aviscera Bioscience, Santa Clara, CA, USA) or CTRP8 (RD172179100, BioVendor, Brno, Czech Republic) (*n* = 9, respectively). The previously imaged area was photographed again at 6, 12, 18, 24, 36, and 48 h after wounding. The surface defect area was assessed at each time point using gimp (Version 2.10.14) and paint.net (Version 4.3.12). Additionally, after performing linear regression analysis of the assessed defect areas at the different time points for each group, the surface defect closure rate was calculated by ratio of slope and y-intercept to represent the relative decrease of the initial surface defect area per hour. Control levels were set to 1 to display fold-increase. Samples stimulated with either 10 ng/mL or 100 ng/mL of the respective CTRP were compared to untreated controls (*n* = 9).

The dependence of CTRP effects on the RXFP1 receptor pathway was assessed by additional scratch assays performed as described above. In addition to stimulation with 100 ng/mL of CTRP1, CTRP6, or CTRP8, HCE cells were also treated with 3 µg/mL of a specific inhibitory anti-RXFP1 antibody (H00059350-M01, Abnova, Taipei, Taiwan) [42] to block any possible RXFP1-dependent stimulation and the defect area was photographed at 0, 6, 12, 24, and 48 h after wounding. The stimulation of HCE cells with 100 ng/mL relaxin-2 (RLN2; #035-62, Phoenix Pharmaceuticals, Burlingame, CA, USA) in the absence or presence of the antibody served as positive control [3] and confirmed inhibition of RXFP1 by the antibody. To determine any effects on ocular surface wound closure by the antibody alone, negative controls with the RXFP1 antibody present or absent were performed (*n* ≥ 9, respectively).

### 3.7. Statistical Analysis

All data are shown as mean ± standard error of mean (SEM). Gaussian distribution was calculated by the Kolmogorov–Smirnov test. After evaluating values for normal distribution, we performed two-way ANOVA statistics in combination with a Tukey’s multiple comparison test to compare the ocular surface defect area at different time points, respectively. Statistical significance of the surface defect closure rate among the groups was analyzed by one-way ANOVA statistics if comparing more than two groups or unpaired t-test if comparing two groups. For the interpretation of the results, Dunnett or Šidák post hoc test was used. Significance was defined at * *p* < 0.05, ** *p* < 0.01 or *** *p* < 0.001. All charts were generated and analyzed statistically with GraphPad Prism (Version 9.4.1; GraphPad Software, San Diego, CA, USA).

## 4. Discussion

Ocular surface wound healing is a highly complex process that involves the synergistic interactions of various growth factors, cytokines, and other hormones that regulate extracellular matrix (ECM) formation, as well as proliferation, migration, and differentiation of epithelial and stromal cells [1,2]. Besides others, RLN2 and INSL3 and their cognate receptors RXFP1 and RXFP2, respectively, facilitate ocular surface wound healing mainly by promoting increased corneal epithelial cell migration [3,4]. Recently, secretory proteins CTRP1, CTRP6, and CTRP8 have been proposed as novel interaction partners of RXFP1 [5,7,10]. This study represents the first report of CTRP1, CTRP6, and CTRP8 at the ocular surface and describes the potent repair function of these three adipokines in an in vitro ocular surface epithelial wound closure model.

We were able to demonstrate CTRP1 and CTRP6 gene expression in human cornea, conjunctiva, lacrimal gland, and nasolacrimal ducts as well as corneal (HCE and hTCEpi), conjunctival (HCjE and IOBA-NHC), and meibomian gland (HMGEC; differentiated and undifferentiated) epithelial cell lines. CTRP8 gene expression was only detected in HCE and IOBA-NHC cells by RT-PCR (Figure 1). Previously, PCR analysis revealed restricted expression of CTRP8 in human lung and testis [9]. However, immunohistochemical results revealed CTRP1, CTRP6, and CTRP8 expression at the protein level in tissues and cell lines of the ocular surface and lacrimal apparatus, including meibomian glands, lacrimal glands, as well as epithelium of the nasolacrimal ducts, cornea, and conjunctiva (Figure 2 and Figure 3). Accordingly, we suggest tissues of the ocular surface and lacrimal apparatus as a new source of CTRP1, CTRP6, and CTRP8. Interestingly, CTRP6 expression in acinar cells of salivary glands and its secretion into the salvia have been described recently [43]. Thus, assuming that CTRP1, CTRP6, and CTRP8 are secreted into the tear fluid as well seems appropriate. Moreover, both CTRP1 and CTRP6 concentrations have already been measured in blood samples and serum levels were elevated in obese subjects [6,20,24,44] as well as in subjects suffering from associated diseases such as metabolic syndrome [45], type 2 diabetes mellitus [22,45,46,47,48,49,50,51], hypertension [52], and coronary artery disease [16,53,54]. Pathogenesis of these diseases is linked to low-grade local tissue and systemic inflammation through dysregulation of pro-inflammatory cytokines, including TNF-α, IL-1β, and IL-6 [55,56,57]. Notably, high CTRP1 and CTRP6 serum levels are also associated with increased TNF-α, IL-1β, and IL-6 blood concentrations [16,17,22,47,49,54,58], and CTRP1 and CTRP6 serum levels are increased in patients suffering from chronic inflammatory diseases, such as rheumatoid arthritis and Kawasaki disease [17,24]. Pro-inflammatory cytokines also play a major role in the initiation of ocular surface wound healing and are up-regulated in the tear fluid of patients with dry eye disease [1,59,60]. When injured, damaged or dead corneal epithelium cells release IL-1α and IL-1β, which cause keratocyte and corneal fibroblast activation, migration of immune cells, and further chemokine release [1,60,61,62]. Exposure to IL-1β and IL-1α directly increased CTRP1 and CTRP6 expression levels in adipose tissue and primary fibroblast-like synoviocytes, respectively [18,24]. Hence, we investigated whether CTRP1, CTRP6, and CTRP8 affect ocular surface epithelial wound closure as well.

Based on our results of CTRP1, CTRP6, and CTRP8 protein expression in HCE and HCjE cell lines (Figure 3), we considered these cell lines appropriate for further epithelial wound closure experiments. Our scratch assays on HCE and HCjE as an in vitro corneal and conjunctival epithelial wound model determined the effects of CTRP1, CTRP6, and CTRP8 on proliferation and especially migration of epithelial cells of the ocular surface. We observed dose-dependent effects of CTRP1, CTRP6, and CTRP8 on corneal but not on conjunctival epithelial wound closure in vitro (Figure 4, Figure 5, Figure 6 and Figure 7). After 6–12 h of treatment with 100 ng/mL recombinant CTRP1, CTRP6, or CTRP8, HCE showed a significant reduction in the remaining wound area compared to the untreated controls. By contrast, treatment with CTRP1, CTRP6, or CTRP8 had no significant effect on the remaining wound area in HCjE. Cell migration is a vital part of ocular surface wound healing and essential to restore integrity of the physiological barrier of the eye [1,60]. Several studies have described enhanced cell proliferation and migration as well as tumor invasion through CTRP1, CTRP6, or CTRP8 [5,7,19,20,21,30,31,63]. For example, CTRP8 promotes cell migration in human glioblastoma cells by activation of the RXFP1 involving the N-terminal “YAAFSVG” peptide motif present in the C1q globular domain of CTRP8 [5]. Close evolutionary relationship and structural similarities of CTRP1 and CTRP6 with CTRP8 suggest that CTRP1 and CTRP6 may potentially interact with RXFP1 as well [5,10]. This is further supported by the fact that CTRP1 shares the same “YAAFSVG” peptide motif as CTRP8 and inhibition of glioblastoma cell proliferation and migration upon selective CTRP1 knockdown [10,21]. On the other hand, CTRP6 has a modified RXFP1 peptide binding motif, which may suggest that CTRP6 acts as a competitive antagonist at the RXFP1 receptor instead [10]. The RLN2-RXFP1 receptor pathway is a known component of ocular surface wound healing. At the ocular surface, binding of RLN2 and ISNL3 to its cognate receptors RXFP1 and RXFP2 promotes moderately increased cell proliferation and markedly increases re-epithelialization mainly through increased cell migration of corneal cells [3,4]. However, even though RXFP1 is expressed in both, HCE as well as HCjE, RLN2 has been shown to increase cell migration only in corneal epithelial cells but not in conjunctival epithelial cells [3]. This is consistent with our findings that CTRP1, CTRP6, and CTRP8 promote exclusive wound closure of corneal epithelial cells and might be due to different biology and downstream signaling in HCjE compared to the HCE cell line. RLN2-RXFP1/2 interaction increases the expression of matrix metalloproteinases (MMPs) and tissue inhibitors of matrix metalloproteinases (TIMPs) which play important roles in wound healing and tissue remodeling homeostasis [1,3]. We therefore focused on possible effects of CTRP1, CTRP6, and CTRP8 on corneal epithelial wound closure facilitated by RXFP1.

To investigate the dependence of CTRP1, CTRP6, and CTRP8 effects on RXFP1 signaling pathways, we designed a competitive RXFP1 binding assay in which a specific RXFP1 receptor antibody blocked CTRP ligand binding to RXFP1 and tested the effect on cell motility in our scratch assays on HCE. RXFP1 expression in HCE has already been described by Hampel et al. [3]. Prior to stimulation, we confirmed RXFP1 expression by immunohistochemical reactivity and antibody-mediated inhibition of RXFP1 was confirmed in scratch assays on HCE treated with 100 ng RLN2 with or without antibody treatment. Consistent with the findings of Hampel et al. and Ferlin et al., stimulation with RLN2 significantly increased the surface defect closure rate in HCE, whereas the presence of the RXFP1 antibody fully abolished the effects of RLN2 stimulation. The antibody alone, however, did not affect surface defect closure (Figure 8) [3,42].

While treatment with CTRP1, CTRP6, or CTRP8 led to a significantly increased surface defect closure rate in HCE, inhibition of RXFP1 influenced the CTRP-treatment effects differently (Figure 9). Similar to RLN2, the effects of CTRP8 on the surface defect closure rate were fully diminished by RXFP1 inhibition, suggesting that CTRP8-induced epithelial wound closure is mediated mainly via CTRP8-RXFP1 interaction.

By contrast, RXFP1 inhibition reduced CTRP1 effects on the surface defect closure rate not significantly (1.41 ± 0.08-fold (CTRP1) vs. 1.29 ± 0.08-fold (CTRP1 + ab); *p* = 0.80). This might indicate that CTRP1-RXFP1 interaction is only partly responsible for the effect of CTRP1 on ocular surface wound closure and additional receptors and signaling pathways are involved that need yet to be investigated. CTRP1 was shown to regulate expression of inflammation-related genes including IL-1β, IL-6, TNF-α in human vascular smooth muscle cells and primary human macrophages [14,15]. IL-6 expression is increased after corneal damage and leads to increased corneal wound closure by proliferation and migration of corneal epithelial cells [64,65,66,67,68,69]. Therefore, RXFP1-independent CTRP1 effects on corneal epithelial wound closure may include increased IL-6 expression. On the contrary, CTRP1 has also shown to decrease inflammatory responses in microglia and after myocardial infarction, indicating that pro- and anti-inflammatory effects of CTRP1 might be tissue and cell type specific [13,58]. Further functional studies regarding the signaling pathway of CTRP1 and its role in ocular surface wound healing are warranted.

The stimulation with CTRP6 increased the surface defect closure rate in HCE significantly, yet inhibition of RXFP1 did not significantly decrease the CTRP6 effects on the surface defect closure rate. However, there was also no significant difference in the surface defect closure rate when compared to the untreated control after RXFP1 inhibition, implying some form of RXFP1 activation by CTRP6. Previous studies have shown that a CTRP6-derived small peptide containing its N-terminal region of the globular C1q domain (“FFAFSVG”) was not able to activate RXFP1 but, instead, could attenuate RXFP1 signaling in the presence of CTRP8 [5,10]. CTRP6 is able to assemble into heterotrimeric formations with CTRP1 but not CTRP8 [6,8,9]. Such heterotrimers may modulate CTRP6 binding and biological activities. Alternatively, soluble CTRP6 can compete with CTRP1/8 for RXFP1 binding, which can affect RXFP1 signaling responses. Because RXFP1 inhibition was not able to significantly attenuate the CTRP6 effects on corneal epithelial wound closure, other signaling pathways might also be involved. Contradictory findings regarding the CTRP6 effects on cell proliferation in oral squamous cell cancer cells further emphasize the complexity of CTRP6 signaling and the need for further investigations [31,32].

Our results suggest a novel role for CTRP1, CTRP6, and CTRP8 at the ocular surface and in corneal epithelial wound closure in vitro by targeting—at least partly—the RXFP1 receptor pathway. Incubation with an antibody to the RXFP1 extracellular domain reduced the CTRP8 effects on corneal epithelial wound closure to control levels, suggesting that RXFP1 signaling mediates CTRP8 effects on corneal wound closure in vitro. However, inhibition of RXFP1 did not fully block CTRP1- and CTRP6-mediated effects on corneal epithelial wound closure. This suggests that, in addition to signaling through RXFP1, CTRP1 and CTRP6 may utilize additional unknown signaling pathways to support corneal wound closure. Our findings are limited as wound closure experiments were only performed in vitro on immortalized cell lines. Future in vivo studies are needed to obtain a more in depth understanding of the function of CTRP1, CTRP6, and CTRP8 at the ocular surface. Many studies have reported a distinct involvement of CTRP1 and CTRP6 in inflammatory processes. For CTRP1, mostly pro-inflammatory properties have been described [14,15,17]. On the other hand, CTRP6 enhances IL-10 expression in macrophages and, through inhibition of the alternative pathway of the complement system, can exert anti-inflammatory properties in rheumatoid arthritis patients [24,25]. Dysregulation of inflammatory processes is critical in the pathogenesis of Sjögren’s syndrome, a common cause of dry eye disease, and anterior uveitis, and it might be connected to CTRP signaling. In corneal wound healing, functional interaction of different cytokines facilitates proliferation, migration, and differentiation of epithelial and stromal cells as well as extra cellular matrix formation to ensure intact recovery. On the other hand, dysregulation of this process can lead to permanent vision impairment through chronic ulcera and corneal fibrosis. Thus, proper ocular surface wound healing represents an interesting approach for pharmacological and therapeutic intervention [1,70]. Both CTRP1 and CTRP6, as well as the RXFP1, a known binding partner of CTRP8, have shown anti-fibrotic properties in various types of tissues [71]. CTRP1 inhibits renal fibrosis and antagonizes angiotensin II induced cardiac fibrosis [12,72]. Interestingly, RXFP1 signaling might be involved in CTRP1 anti-fibrotic effects, as recent studies revealed angiotensin II type 1/2 receptor (AT_1/2_)-RXFP1 interactions that directly or allosterically interfere with ligand signal transduction in myofibroblasts [73]. CTRP6 inhibits TGF-β-mediated fibrosis in renal, cardiac, and dermal fibroblasts [26,27,28,29]. Notably, similar effects have been described for RLN2 in those cell types [74,75,76]. Treatment with protease-resistant RXFP1-activating peptides may be an attractive approach to antagonize TGF-β-induced excessive fibroblast activation and reduce corneal fibrosis and scarring [2,71].

## 5. Conclusions

In summary, our present study is the first to demonstrate that CTRP1, CTRP6, and CTRP8 are present at the ocular surface and are involved in corneal wound closure in vitro. These findings further suggest an involvement of RXFP1 signaling in CTRP1/6/8-mediated wound closure at the ocular surface and provide evidence of additional yet unidentified signaling pathways for CTRP1 and CTRP6, supporting this new CTRP1/6/8 biology. Further research on the function of CTRP1, CTRP6, and CTRP8 and their role in ocular surface pathologies may provide future opportunities for drug discovery to treat corneal wounds.

## Figures and Tables

**Figure 1 ijms-24-06839-f001:**
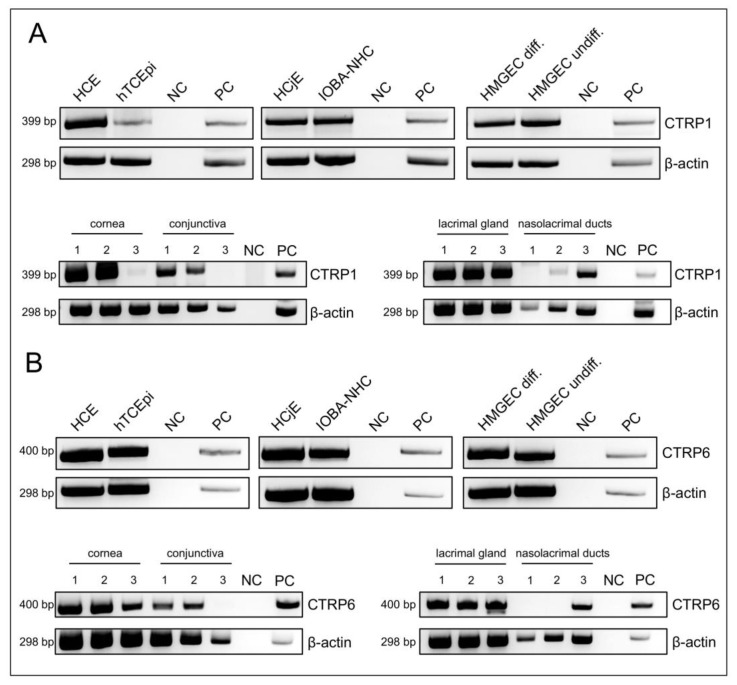
**CTRP1 and CTRP6 gene expression in human tissues and cell lines by RT-PCR.** (**A**) Gene expression of CTRP1. (**B**) Gene expression of CTRP6. The gene expression is shown by the black bands. Expression of β-actin confirms the presence of tested cDNA. Human corneal epithelial cells (HCE), telomerase-immortalized human corneal epithelial cells (hTCEpi), human conjunctival epithelial cells (HCjE), IOBA-normal human conjunctiva (IOBA-NHC), human meibomian gland epithelial cells (HMGEC; differentiated and undifferentiated). Positive control (PC) is liver for CTRP1 and placenta for CTRP6. Negative control (NC) contains no template cDNA. Pictures are representative for at least three individual experiments for each tissue and cell line.

**Figure 2 ijms-24-06839-f002:**
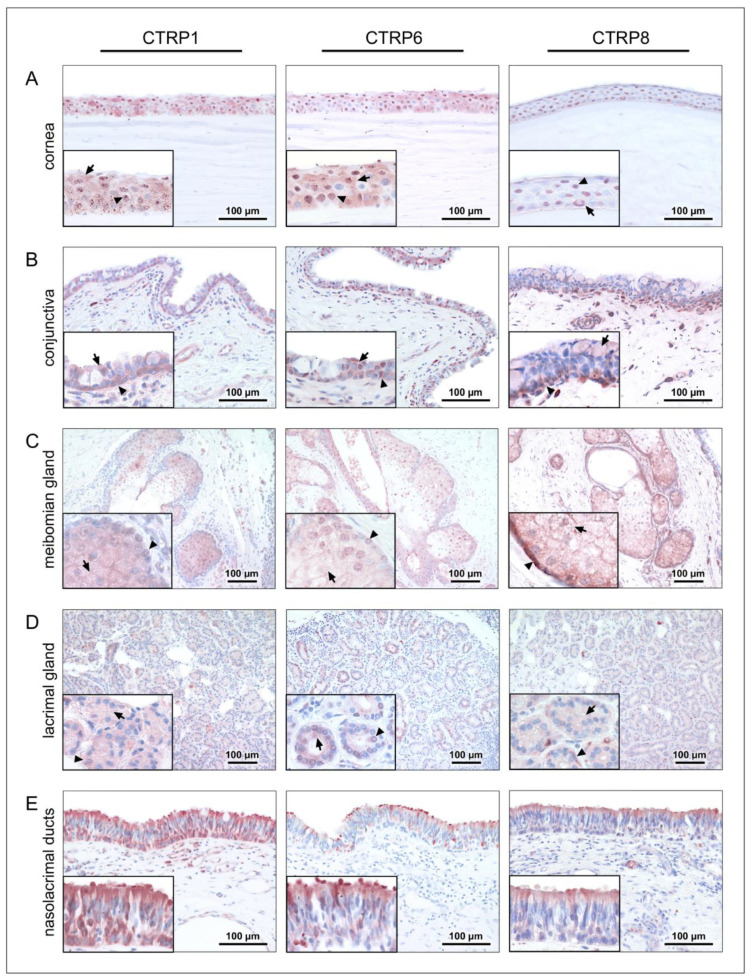
**Immunohistochemical detection of CTRP1, CTRP6, and CTRP8 in tissues of the human lacrimal apparatus and ocular surface.** The antibody reaction can be seen by the intracellular red reactivity. Pictures shown are representative for human cornea (**A**), conjunctiva (**B**), meibomian gland of upper eye lid (**C**), lacrimal glands (**D**), and nasolacrimal ducts (**E**). Inlays show higher magnification. Nuclei are counterstained with hemalum (blue). Arrows and arrowheads accentuate the reactivity localization. For each tissue, at least three individual sections were stained.

**Figure 3 ijms-24-06839-f003:**
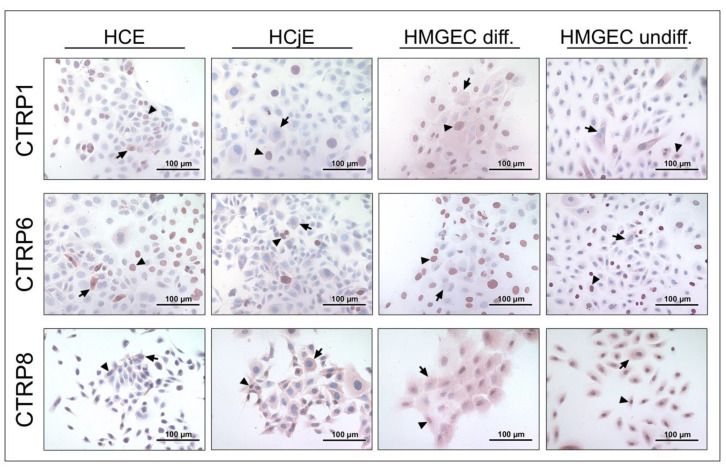
**Immunohistochemical detection of CTRP1, CTRP6, and CTRP8 in cell lines of the human lacrimal apparatus and ocular surface.** The antibody reaction can be seen by the intracellular red reactivity. Pictures shown are representative for human corneal epithelial cells (HCE), human conjunctival epithelial cells (HCjE), differentiated human meibomian gland epithelial cells (HMGEC), and undifferentiated HMGEC. Nuclei are counterstained with hemalum (blue). Arrows and arrowheads accentuate the reactivity localization. For each cell line, three individual experiments were performed.

**Figure 4 ijms-24-06839-f004:**
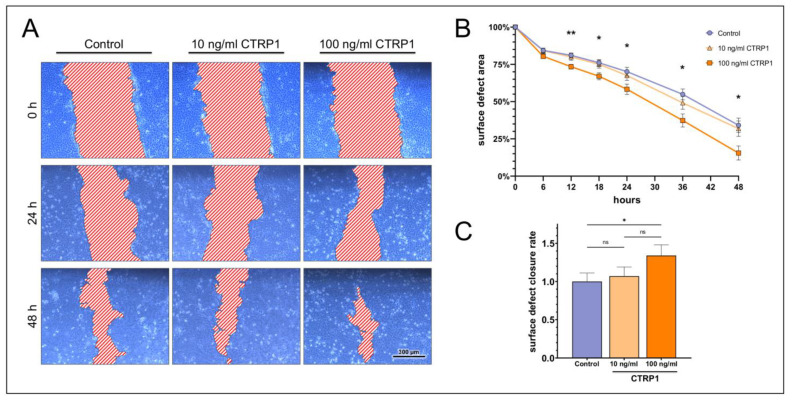
**Scratch assays show the dose-dependent impact of CTRP1 on surface defect closure in human corneal epithelial cells (HCE).** (**A**) After performing the initial scratch, HCE cells were stimulated with 0, 10, or 100 ng/mL recombinant human CTRP1 for 48 h. The surface defect was measured in a representative segment of a scratch (*n* = 9) at different time points and compared to untreated cells (control). (**B**) Relative surface defect area ± SEM over time. The graph shows a significant decrease in the relative surface defect area in cells treated with 100 ng/mL CTRP1 compared to the control (two-way ANOVA; * *p* < 0.05; ** *p* < 0.01). Treatment with 10 ng/mL CTRP1 had no significant effect on the surface defect area when compared to the control (two-way ANOVA; *p* > 0.05). (**C**) Fold-increase in the surface defect closure rate. Bars represent mean ± SEM. 100 ng/mL CTRP1 increased the relative surface defect closure rate significantly compared to the control (one-way ANOVA; * *p* < 0.05), while 10 ng/mL CTRP1 was ineffective (one-way ANOVA; ns = not significant, *p* > 0.05). The difference in the surface defect closure rate between HCE cells treated with 10 ng/mL and 100 ng/mL CTRP1 was not significant (one-way ANOVA; ns = not significant, *p* > 0.05).

**Figure 5 ijms-24-06839-f005:**
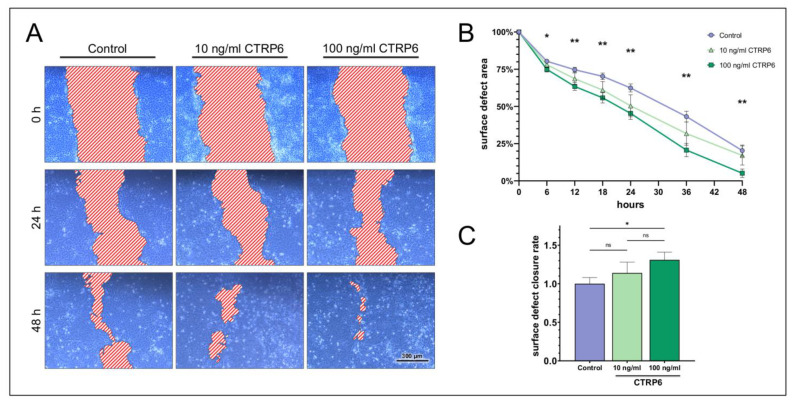
**Scratch assays show the dose-dependent impact of CTRP6 on surface defect closure in human corneal epithelial cells (HCE).** (**A**) After performing the initial scratch, HCE cells were stimulated with 0, 10, or 100 ng/mL recombinant human CTRP6 for 48 h. The surface defect was measured in a representative segment of a scratch (*n* = 9) at different time points and compared to untreated cells (control). (**B**) Relative surface defect area ± SEM over time. The graph shows a significant decrease in the relative surface defect area in cells treated with 100 ng/mL CTRP6 compared to the control (two-way ANOVA; * *p* < 0.05; ** *p* < 0.01). Treatment with 10 ng/mL CTRP6 had no significant effect on the surface defect area when compared to the control (two-way ANOVA; *p* > 0.05). (**C**) Fold-increase in the surface defect closure rate. Bars represent mean ± SEM. 100 ng/mL CTRP6 increased the relative surface defect closure rate significantly compared to the control (one-way ANOVA; * *p* < 0.05), while 10 ng/mL CTRP6 was ineffective (one-way ANOVA; ns = not significant, *p* > 0.05). The difference in the surface defect closure rate between HCE cells treated with 10 ng/mL and 100 ng/mL CTRP6 was not significant (one-way ANOVA; ns = not significant, *p* > 0.05).

**Figure 6 ijms-24-06839-f006:**
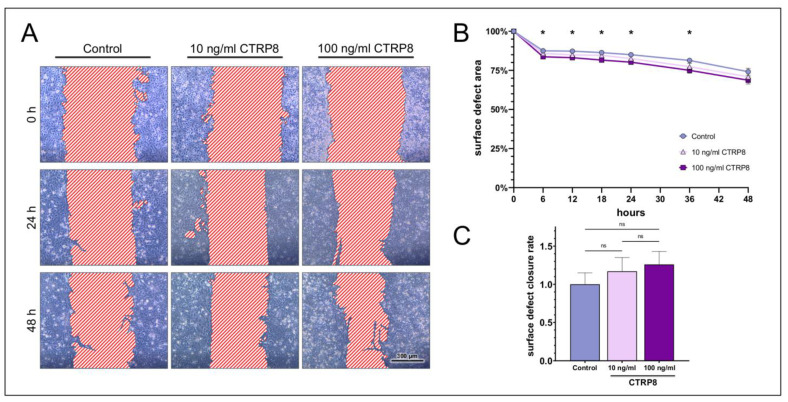
**Scratch assays show the dose-dependent impact of CTRP8 on surface defect closure in human corneal epithelial cells (HCE).** (**A**) After performing the initial scratch, HCE cells were stimulated with 0, 10, or 100 ng/mL recombinant human CTRP8 for 48 h. The surface defect was measured in a representative segment of a scratch (*n* = 9) at different time points and compared to untreated cells (control). (**B**) Relative surface defect area ± SEM over time. The graph shows a significant decrease in the relative surface defect area in cells treated with 100 ng/mL CTRP8 compared to the control at 6 h but lost significance at 48 h (two-way ANOVA; * *p* < 0.05). Treatment with 10 ng/mL CTRP8 had no significant effect on the surface defect area when compared to the control (two-way ANOVA; *p* > 0.05). (**C**) Fold-increase in the surface defect closure rate. Bars represent mean ± SEM. 10 ng/mL and 100 ng/mL CTRP8 increased the relative surface defect closure rate compared to the control but not significantly (one-way ANOVA; ns = not significant, > 0.05). The difference in the surface defect closure rate between HCE cells treated with 10 ng/mL and 100 ng/mL CTRP8 was also not significant (one-way ANOVA; ns = not significant, *p* > 0.05).

**Figure 7 ijms-24-06839-f007:**
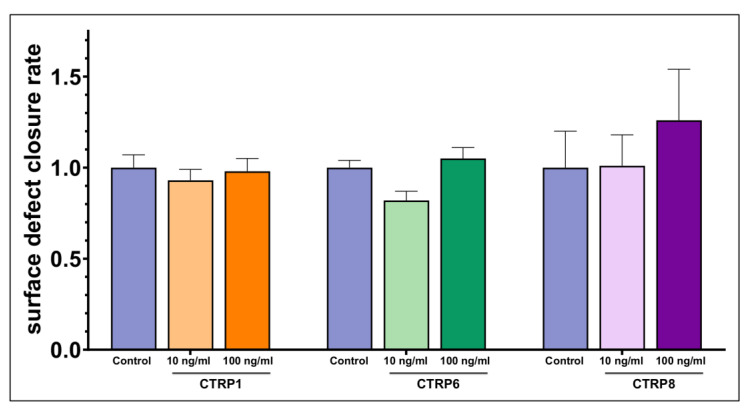
**Scratch assays show the dose-dependent impact of CTRP1, CTRP6, and CTRP8 on surface defect closure in human conjunctival epithelial cells (HCjE).** Fold-increase in the surface defect closure rate. Bars represent mean ± SEM. Analog to HCE, scratch assays were performed on HCjE cell line. Neither stimulation with 10 ng/mL nor 100 ng/mL CTRP1, CTRP6 or CTRP8 did increase the relative surface defect closure rate significantly when compared to the control (one-way ANOVA; *p* > 0.05).

**Figure 8 ijms-24-06839-f008:**
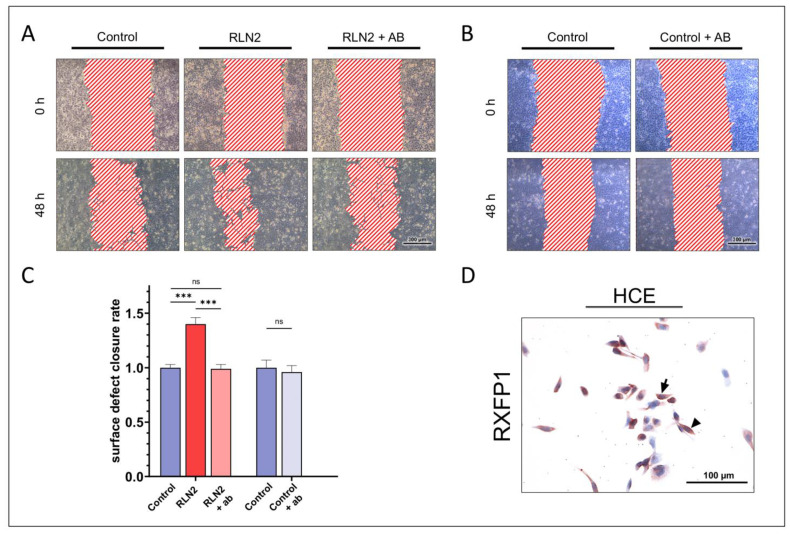
**Scratch assays confirm inhibition of RXFP1.** (**A**,**B**) show representative scratch assays in human corneal epithelial cells (HCE). Epithelial wound closure in cells stimulated with 100 ng/mL Relaxin-2 (RLN2, positive control) (**A**) or untreated cells (negative control) (**B**) was observed, respectively, with or without the presence of 3 mg/mL of a specific inhibitory anti-RXFP1 antibody (ab). The surface defect was measured in a representative segment of a scratch (*n* = 9) at different time points and compared to untreated HCE cells (control). (**C**) Fold-increase in the surface defect closure rate. Bars represent mean ± SEM. RLN2 significantly increased the surface defect closure rate compared to the untreated control (one-way ANOVA; *** *p* < 0.001). The presence of the inhibitory anti-RXFP1 antibody significantly reduced the effect of RLN2 on the surface defect closure rate (one-way ANOVA; *** *p* < 0.001). There were no significant differences in the surface defect closure rate between the untreated control and HCE that were treated with RLN2 but also underwent RXFP1 inhibition (one-way ANOVA; ns = not significant, *p* > 0.05). The antibody alone did not significantly alter the surface defect closure rate (unpaired t-test; ns = not significant, *p* > 0.05). (**D**) RXFP1 protein expression was confirmed prior to scratch assays in HCE cells via immunohistochemistry. The antibody reaction can be seen by the red reactivity. Nuclei are counterstained with hemalum (blue). Arrows and arrowheads accentuate the reactivity localization. Three individual experiments were performed.

**Figure 9 ijms-24-06839-f009:**
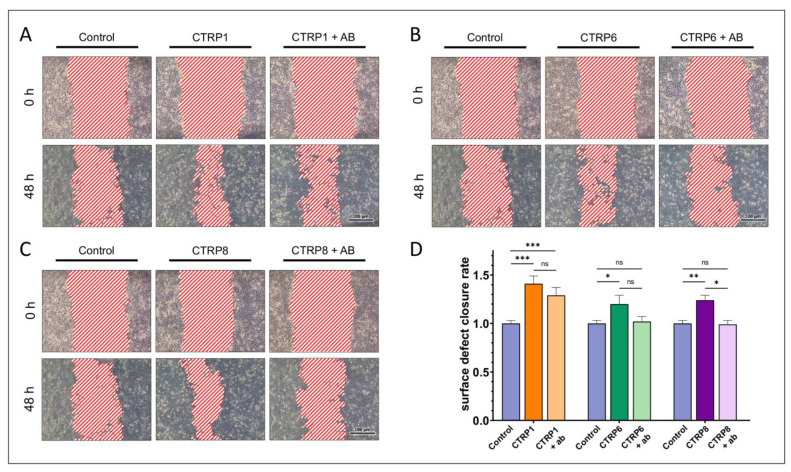
**Scratch assays show the dependence of CTRP1, CTRP6, and CTRP8 effects on the RXFP1 receptor pathway.** (**A**–**C**) show representative scratch assays in human corneal epithelial cells (HCE). Cells were stimulated with 100 ng/mL CTRP1 (**A**), CTRP6 (**B**), or CTRP8 (**C**) for 48 h with or without the presence of 3 mg/mL of a specific inhibitory anti-RXFP1 antibody (ab). The surface defect was measured in a representative segment of a scratch (*n* ≥ 9) at different time points and compared to untreated HCE cells (control). (**D**) Fold-increase in the surface defect closure rate. Bars represent mean ± SEM. All three tested peptides significantly increased the surface defect closure rate compared to the untreated control (one-way ANOVA; * *p* < 0.05; ** *p* < 0.01; *** *p* < 0.001). The presence of the inhibitory anti-RXFP1 antibody significantly reduced the effect of CTRP8 on the surface defect closure rate (one-way ANOVA; * *p* < 0.05). There were no significant differences in the surface defect closure rate between the untreated control and HCE that were treated with CTRP8 but also underwent RXFP1 inhibition (one-way ANOVA; ns = not significant, *p* > 0.05). Inhibition of RXFP1 decreased the surface defect closure rate in CTRP6-treated HCE, but not significantly (one-way ANOVA; ns, *p* > 0.05). However, there was also no significant difference in the closure rate of HCE simultaneously treated with CTRP6 and the anti-RXFP1 antibody compared to the untreated control (one-way ANOVA; ns, *p* > 0.05). CTRP1 effects on the surface defect closure rate in HCE were barely affected by RXFP1 inhibition (one-way ANOVA; ns = not significant, *p* > 0.05) and stayed significantly increased compared to the control (one-way ANOVA; * *p* < 0.05; ** *p* < 0.01; *** *p* < 0.001).

## Data Availability

The data presented in this study are available on request from the corresponding author.

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
