# Peer review of "C1q/TNF-Related Proteins 1, 6 and 8 Are Involved in Corneal Epithelial Wound Closure by Targeting Relaxin Receptor RXFP1 In Vitro"

_ijms, 2023, doi:10.3390/ijms24076839_

Round 1

Reviewer 1 Report

The article is well written and can be accepted in present form.

Reviewer 2 Report

This study explored the role of CTRP1, CTRP6, and CTRP8 in wound closure at the ocular surface by targeting the relaxin receptor RXFP1. The research design is rigorous and the results are clear. However, there are some problems needing to be addressed. Details are listed below.

Abstract:

1.        Abbreviations are not recommended in the abstract. The abbreviations appearing for the first time should be expressed with full name followed by the abbreviation, and the form should be consistent thereafter. (Line 21)

2.        Not all differences are significant about the “dependence on RXFP1 receptor pathway”, the conclusions that CTRP1 and CTRP6 are involved in corneal wound closure by targeting relaxin receptor RXFP1 seem too arbitrary. (Line 38-41)

Key words:

3.        Too many key words.

Introduction:

4.        “ECM” in line 92 appears for the first time but is not expressed in full name and is not defined until Discussion (Line 446).

5.        The significance of the study is not specific enough, and the objective should be stated in detail (Line 95-101.)

Materials and methods:

6.        Please clarify how to calculate the surface defect closure rate (Line 210-224.)

7.        More details about dose-dependent and dependence on RXFP1 receptor pathway methods should be added in this part. (Line 210-234.)

Results and discussion:

8.        The result should not include the description of methods (Line 321-327)

9.        The stimulation of HCE cells with 100 ng/ml CTRP8 lead to a significant decrease in remaining surface defect area at 6, 12, 24 and 36 hours of stimulation but was not significant between groups at 48 hours (Figure 6). (Line 360-361) But in Figure 6 legend, it did not clarify the change at different time points. (Line 370-372)

10.     “(HCE)” s appeared 4 times. The abbreviations appearing for the first time should be interpreted accordingly, and the form should be consistent thereafter. (Line 254, 257, 299, 391)

11.     The interpretation of the results should not appear in Result, but in Discussion. (Line 429-430)

12.     Same, please check all abbreviation issues (Line 486, 493)

Conclusion:

13.     Not every comparison difference is significant about the “Dependence on RXFP1 Receptor Pathway”, the conclusions that CTRP1 and CTRP6 are involved in corneal wound closure by targeting relaxin receptor RXFP1 seem too arbitrary. (Line 595-597)

References:

14.     Many self-citations by the authors. NO. 3, 4, 7, 10, 33, 38-41. Please check the relevance and cite the most relevant ones.

Reviewer 3 Report

The findings obtained in this study, in which the well-planned and applied techniques are explained in detail, are presented beautifully with graphics and pictures. It is important to study that the expression of CTRP1, CTRP6 and CTRP8 at the ocular surface and their role in ocular surface wound healing have been investigated.

Reviewer 4 Report

Ten years ago, the expression of the insulin-like peptide hormone relaxin 2 (RLN2) on the ocular surface of the human eye and its role in promoting wound healing has been investigated in mouse models. Despite earlier studies finding RLN2 expression in the retina, recent research has found little evidence of RLN expression on the mouse ocular surface. The mode of action of RLN/RXFP axis in ocular surface wound healing is still not well understood. Knock-out mice for the relaxin gene (Rln-/-) showed reduced cornea thickness and altered density of endothelial cells, suggesting that the RLN/RXFP axis plays a role in maintaining cornea integrity. Recent studies have also found that C1q-tumour necrosis factor-related protein 8 (CTRP8) is a novel interaction partner of the relaxin receptor RXFP1. Therefore, the current study investigates the effects of CTRP 1, 6, 8 as interactors of RXFP1 on ocular surface wound healing. The authors found that human cornea cell lines expressed abundant mRNA for CTRP 1 and 6, but less for CTRP8. Proteins levels of CTRP 1, 6, 8 were detected in human cornea sections and exposure to recombinant human CTRP 1, 6, 8 proteins promoted wound healing in vitro, which was blocked by the RXFP1 neutralization antibody.

The study has several limitations, including a lack of in vivo experiments and reliance on single experiments, which reduces the rigor of the presented data.

Major:

1. In addition to RT-PCR and IHC, the presence of CTRP 1, 6, 8 protein levels should be validated by immunoblotting to show expected protein bands.

2. Although CTRP8 was shown interacts with RXFP1. Further evidence is needed to support the interaction between CTRP 1, 6, and RXFP1.

3. The difference in potency of CTRP 8 (100ng) in promoting wound healing compared to CTRP 6 needs to be explained (figure 4,5,6,and 9D).

Minor:

Please indicate the scale bar for images 2, 3, 4, 5, 6, and 8.

Reviewer 5 Report

Hagen Fabian Nicolaus and coworkers reported the C1q/tumor necrosis factor-related proteins(CTRP 1, 6, 8) effects of recombinant CTRP1, CTRP6 and CTRP8 on cell proliferation and migration in human tissue/cell samples. Further, they have proved the hypothesis from the Wound Closure Assay (Scratch Assay) for the proteins presented and RXFP1 mechanism. Though most of the experiments are in vitro on cell lines, the study advances the understanding of the wound healing of ocular surface injuries.   

Methods are presented in a detailed manner and the conclusions are supported by the data presented in the article.

This work is very interesting to the readers of IJMS and Deserved to be published after addressing the following comments.

1.     All figures provided are low resolution especially reading those scale bars extremely difficult. Please maintain the same quality you provided in the supplementary files in the final version. The qualitative aspects of these images are crucial for the interpretation of the data for the reader.

2.     Figure 6C should have statistics.?

3.     Figure 7 Statistics are completely missing. Specify comparisons.

4.     “By contrast, treatment with CTRP1, CTRP6 or CTRP8 had no significant effect on the remaining wound area in human conjunctival epithelial cells (HCjE).”

Shed light on why with more details.

5.     I suggest that authors may consider statistics within the groups because control and 10ng/ml effect is almost similar but 100ng effect seems higher in certain cases. In that case, the legend should specify x vs y specifically. If there is no difference, mention that too.

6.     Please rewrite figure legends, and specify which test used and/or ANOVA between each group for better understanding for the reader. 

Round 2

Reviewer 4 Report

Thanks for the Authors providing improvements on the revised manuscript. However, the main concerns about the rigor of data, and missing evidence supporting the major conclusion are not even addressed. 

1> The article title is kind of misleading: "C1q/TNF-related proteins 1, 6 and 8 are involved in wound closure at the ocular surface by targeting relaxin receptor RXFP1"  Actually the wound closure only tested with human corneal epithelial cells, not in vivo ocular surface.

2> Again, in addition to PCR and HC, specific bands of each CTRPs should be confirmed by western blots. Although HC also detects protein levels, it can not exclude nonspecific signals from off-targets, which can be clarified by SDS-PAGE gel electrophoresis and western-blots

Reviewer 5 Report

The manuscript was revised carefully as per my request, and additional data was included in the suppli files. I have no additional comments. The current form is acceptable. 
